# A Protocol for the Acquisition of Comprehensive Proteomics Data from Single Cases Using Formalin-Fixed Paraffin Embedded Sections

**DOI:** 10.3390/mps5040057

**Published:** 2022-07-10

**Authors:** Mitchell Acland, Parul Mittal, Georgia Arentz, Fergus Whitehead, Peter Hoffmann, Manuela Klingler-Hoffmann, Martin K. Oehler

**Affiliations:** 1Adelaide Proteomics Centre, School of Biological Sciences, The University of Adelaide, Adelaide, SA 5005, Australia; mitch.acland@gmail.com (M.A.); georgia.arentz@gmail.com (G.A.); 2Clinical & Health Science, Mawson Lakes Campus, University of South Australia, Adelaide, SA 5095, Australia; parul.mittal@unisa.edu.au; 3Clinpath Pathology, 21 James Congdon Drive, Mile End, Adelaide, SA 5031, Australia; fwhitehead@clinpath.com.au; 4Department of Gynaecological Oncology, Royal Adelaide Hospital, North Terrace, Adelaide, SA 5000, Australia; 5Robinson Research Institute, Discipline of Obstetrics and Gynaecology, Adelaide Medical School, The University of Adelaide, Adelaide, SA 5000, Australia

**Keywords:** serous endometrial carcinoma, high grade serous ovarian carcinoma, endometrial intraepithelial carcinoma, serous tubal intraepithelial carcinoma, proteomics, laser capture microdissection, MALDI mass spectrometry imaging, LC-MS/MS

## Abstract

The molecular analysis of small or rare patient tissue samples is challenging and often limited by available technologies and resources, such as reliable antibodies against a protein of interest. Although targeted approaches provide some insight, here, we describe the workflow of two complementary mass spectrometry approaches, which provide a more comprehensive and non-biased analysis of the molecular features of the tissue of interest. Matrix-assisted laser desorption/ionization (MALDI) mass spectrometry imaging (MSI) generates spatial intensity maps of molecular features, which can be easily correlated with histology. Additionally, liquid chromatography tandem mass spectrometry (LC-MS/MS) can identify and quantify proteins of interest from a consecutive section of the same tissue. Here, we present data from concurrent precancerous lesions from the endometrium and fallopian tube of a single patient. Using this complementary approach, we monitored the abundance of hundreds of proteins within the precancerous and neighboring healthy regions. The method described here represents a useful tool to maximize the number of molecular data acquired from small sample sizes or even from a single case. Our initial data are indicative of a migratory phenotype in these lesions and warrant further research into their malignant capabilities.

## 1. Introduction

Matrix-assisted laser desorption/ionization (MALDI) mass spectrometry imaging (MSI) is a powerful technique for providing spatial information about the molecular environment of a tissue section without homogenization. This provides greater detail than traditional immunohistochemistry approaches and complements pathologist annotation of regions of interest within a tissue.

This can be complemented by liquid chromatography tandem mass spectrometry (LC-MS/MS). This technique can be applied to simultaneously identify hundreds of proteins in a single sample providing a temporal snapshot into its molecular features, as recently illustrated by the comprehensive analysis of glioblastoma tissue [1]. As proteins represent the functional molecules of the cell, this technique has the potential to discover the state of the cell, including its malignant and migratory capabilities. This complements MALDI MSI analysis by providing confident protein identification and the potential to provide relative quantification of protein abundance between samples.

Once a tissue has been surgically removed, it is commonly preserved via formalin fixation and paraffin embedding (FFPE) for storage and analysis by a pathologist. This technique holds the advantage of keeping tissues relatively intact for decades, facilitating retrospective studies and the acquisition of large sample numbers for molecular investigations. However, the formalin fixation process induces protein crosslinks that until recently were incompatible with downstream proteomics analyses, including MALDI and LC-MS/MS. We and others have established an antigen retrieval method that can be used in combination with these techniques to access proteins from FFPE tissues [2].

To demonstrate the potential of these complementary techniques, we applied them to the investigation of precancerous lesions (PLs) of the endometrium and fallopian tube derived from a single patient. Recent advances in the understanding of where serous cancers of the ovary and endometrium originate from [3,4] and the malignant capabilities of their PLs [4,5,6] highlight their relevance in cancer progression and merit deeper molecular investigation.

In serous endometrial cancer (SEC), its precursor, endometrial intraepithelial carcinoma (EIC) [7], is often accompanied by distant micrometastases in the absence of developed SEC [4,5], while high-grade serous ovarian carcinoma (HGSOC) has been identified as originating from serous tubal intraepithelial carcinomas (STIC) located in the fallopian tube [8,9,10,11,12]. These PLs are predicted to migrate to the ovary before they establish a primary tumor. The fact that both PLs hold the potential to migrate to secondary locations before establishing primary tumors suggests that they possess undiscovered molecular drivers of a migratory phenotype. DNA sequencing for *TP53* using precursor lesions from patients with pathogenic variants of BRCA1/2 was able to discover two types of p53 signatures, correlated with either a low or high risk of progression to STIC and ovarian cancer [13]. However, the presentation of HGSOC with the concurrent presence or absence of STIC results in a comparable disease progression [14].

Genomic studies [15] and mouse models [16,17] have been instrumental in establishing the connection between STIC and HGSOC. However, they have not revealed the molecular events that underpin the observed migration of these supposedly pre-malignant cells. Second Harmonic Generation Microscopy and mass spectrometry have been successfully used to evaluate changes in collagens in STIC lesions and concurrent High Grade Serous Ovarian Cancer [6]. These results further imply that mass spectrometry methods, including LC-MS/MS and MALDI MSI, hold the potential to expand our understanding of these early malignant events.

Here, we use a complementary mass spectrometry approach to investigate PLs, which were first identified using disease-specific antibodies and annotated by an experienced pathologist (Prof. F. Whitehead). We use the pathologist’s guidance to identify small, precancerous regions of interest and investigate them with MALDI MSI. In addition, using laser capture microdissection (LCM), we demonstrate how small regions of interest can be isolated for in-depth analysis via LC-MS/MS. We also demonstrate how the use of antigen retrieval and LCM can be used to facilitate MALDI MSI and LC-MS/MS analysis of FFPE tissue to provide complementary spatial and in-depth proteomic information about small tissue regions of interest.

Here, we provide a workflow through which small cancerous samples can be investigated via complementary MALDI MSI and LC-MS/MS approaches (Figure 1). This is facilitated through antigen retrieval, precise microdissection, and well-established sample-preparation methods coupled with high-sensitivity mass spectrometry. We have successfully performed this protocol on PLs from the fallopian tube and endometrium of a single patient providing spatially defined and detailed proteomics information on these biologically relevant tissues.

## 2. Experimental Design

### 2.1. Materials

Acetonitrile, LC-MS/MS grade (1000302500, ACN, Merck, Darmstadt, Germany)A-cyano-4-hydroxycinnamic acid (201344, HCCA, Bruker Daltonics, Bremen, Germany)Ammonium bicarbonate (103025E, Merck, Feltham Middlesex, UK)Citric acid monohydrate (C0706, Sigma-Aldrich, Tokyo, Japan)Dithiothreitol (D9163, DTT, Sigma-Aldrich, Berlington, Massachusetts, USA)Eosin (HT110332, Sigma-Aldrich, Hamburg, Germany)ESI-L low concentration tuning mix (G1969-85000, Agilent, Santa Clara, California, USA)Ethanol AR grade (4.10230.2511, Merck, Bayswater, Victoria, Australia)Ethanol, LC-MS/MS grade (1117272500, Merck, Bayswater, Victoria, Australia)Formic acid, LC-MS/MS grade (21909098, FA, Sigma-Aldrich, Hamburg, Germany)Iodoacetamide (RPN6302OL/AG, IAA, GE Healthcare, Danderyd, Sweden)Isopropanol AR grade (8187661000, Merck, Darmstadt, Germany)Mayer’s hematoxylin (GH5232, Sigma-Aldrich, Hamburg, Germany)Methanol LC-MS/MS grade (1.06018.2500, Merck, Darmstadt Germany)Milli-Q^®^ ultrahigh purity water (D11951, ≥18.2 MΩ, Barnstead International, Dubuque, IA, USA)Mounting medium (PER40000, Medite, Burgdorf, Germany)Neutral buffered formalin (HT501, Sigma-Aldrich, Hamburg, Germany)Sodium hydroxide (1.06498.0500, NaOH, Merck, Darmstadt Germany)Trifluoroacetic acid (1.08262.0100, TFA, Merck, Darmstadt Germany)Trypsin Gold (V5280, Promega, Maddison, Wisconsin, USA)Trypsin sequencing grade (V51111, Promega, Maddison, Wisconsin, USA)Xylene (XA003, Chem-Supply, Gillmann, South Australia, Australia)Urea (1084870500, Merck, Darmstadt, Germany)

### 2.2. Equipment

Centrifuge 5810R (Eppendorf, Hamburg, Germany)CyberScan PC 300 pH Meter (Eutech Instruments, Singapore, Singapore)Data analysis software (e.g., MaxQuant 1.5.2.8)Dry block heater (Ratek, Boronia, Victoria, Australia)Flexcontrol v3.4, flexImaging v4.0 and flexanalysis v4.0 software (Bruker Daltonics, Bremen, Germany)Glass Coplin Jar (H441, ProSciTech, Kirwan, Queensland, Australia)High performance liquid chromatography (HPLC) vials (6820.0029, Dionex, Amsterdam, The Netherlands)ImagePrep (Bruker Daltonics, Bremen, Germany)Incubator at 37 °C (Boekel Scientific, Trevose, PA, USA)Indium-tin-oxide (ITO) slides (237001, Bruker Daltonics, Bremen, Germany)Leica EG 114OH embedder (Leica Biosystems, Mount Waverly, Victoria, Australia)Leica TP 1020 processor (Leica Biosystems, Mount Waverly, Victoria, Australia)Leica Zeiss Laser capture microdissection system (Leica Microsystems, Wetzlar, Germany)Mass spectrometer (e.g., Maxis Impact II QTOF (Bruker Daltonics, Bremen, Germany)Mass spectrometric data acquisition software [e.g., QTOF control (version 3.4) and hystar (version 3.2)]Microm HM 325 microtome (Zeiss, Niedersachsen, Germany)Microwave (LG 700 W MS19496, LG, Jinzhou, China)MTP slide adaptor II (Bruker Daltonics, Bremen, Germany)NanoDrop 2000 (Thermo-Fisher Scientific, Minneapolis, MN, USA)Nanozoomer (Hamamatsu, Hamamatsu City, China)nano-HPLC Ultimate 3000 RS system (Dionex, Amsterdam, The Netherlands)Polyethylene naphthalate (PEN) membrane slides (11505158, MicroDissect, Herborn, Germany)Thermomixer (Eppendorf, Hamburg, Germany)Tipp-Ex (water-based white out, Winc, Wingfield, South Australia, Australia).Scanner (CanoScan 5600 F, Canon, Phra Nakhon Si Ayutthaya, Thailand).SpeedVac concentrator (Savant SVC 100, Thermo Scientific, Minneapolis, MN, USA)SCiLS lab 2016b (Bruker Daltonics, Germany)Superfrost Plus microscopic slides (6.700 125, Thermo scientific, Minneapolis, Minnesota, Germany)Ultraflextreme MALDI TOF (Bruker Daltonics, Bremen, Germany)Ultra-low −80 °C freezer (SANYO, Tokyo, Japan)Vivacon ultrafiltration spin columns (VS0101, Sartorius Vivacon 500, 10,000 MWCO HY, Vivacon, Cologne, Germany).ZipTip C18 (ZTC18M096, Millipore, Burlington, MA, USA)

## 3. Procedure

### 3.1. Tissue Collection

Collect the tissue(s) of interest and snap freeze it in liquid nitrogen immediately after the surgery.

1.In our study, tissues were collected after a total abdominal hysterectomy and bilateral salpingo-oophorectomy was performed on a patient with endometrial hyperplasia at the Royal Adelaide Hospital, Adelaide, Australia, with written informed consent from the patient. The study was approved by the ethics committee of the Royal Adelaide Hospital.2.Process the tissue(s) using the standard procedure [19]. Briefly, fix the tissue(s) in 10% (*v*/*v*) neutral buffered formalin overnight at 4 °C, followed by washing with milli-Q^®^ water and storing in 70% (*v*/*v*) ethanol before processing them with a Leica TP 1020 processor. Embed the tissue(s) in paraffin using a Leica EG 114OH embedder. In our study, tissues were processed and embedded by histology services at the University of Adelaide, Adelaide, Australia. Critical step: Formalin fixation time depends on the tissue size; on average, formalin penetrates tissue at a rate of ~1 mm/h [20].3.Section the FFPE block(s) using Microm HM 325 microtome at 4 µm thickness and water bath mount at 38 °C onto PEN membrane slides for LC-MS/MS, ITO slides for MALDI MSI and superfrost slides for staining purposes.4.Dry the slide(s) at 37 °C for 1 h, followed by overnight drying at room temperature.5.For pathological identification, stain the tissue(s) using H&E or antibody using standard protocols [19]. Here, the identification of PLs (Figure 2) was performed by an experienced pathologist, Prof. Whitehead, using H&E and antibody-stained tissue(s) for P53 and M1B1 (data not shown). Areas of STIC, EIC, and adjacent healthy epithelium were annotated as shown in Appendix A.

### 3.2. Laser Capture Microdissection and Protein Extraction for LC-MS/MS

6.Place the PEN membrane slide(s) on a heating block at 60 °C for 5 min, with the tissue facing up.7.Remove paraffin by dipping the slide(s) in xylene for 90 sec followed by 2 min incubation in 100% ethanol (repeat twice) and 5 min incubation in Milli-Q^®^ water (repeat twice), and then stain the slide(s) with hematoxylin for 20 sec or until the tissue appears purple. Critical step: Staining time depends upon the concentration/age of the solution and/or thickness of the tissue section.8.Destain the slide in Milli-Q^®^ water and then with 70% (*v*/*v*) ethanol for 1 min.9.Let the slide dry at room temperature.10.Load the stained slide and compatible LCM tubes into the Leica Zeiss LCM system. Critical step: To avoid tissue sticking to the side, 5 µL of 10 mM citric acid buffer is added to the cap of an LCM tube11.Carefully cut out the regions of interest(s) (ROI) and ensure that each ROI has been collected into the cap of separate tubes. The tissue before and after LCM dissection of the tissue is shown in Appendix A.12.Briefly, centrifuge the tubes and make sure that the tissue region(s) are in the buffer and at the bottom of the tube. Pause step: tissue(s) can be stored for months at −80 °C13.To retrieve the proteins, add 200 µL of 10 mM citric acid and boil the tubes for 100 min at 98 °C on a thermomixer.14.Let the tubes cool at room temperature and centrifuge briefly.15.Protein concentration can be estimated at this point using Bradford assay, EZQ assay, nanodrop, or tryptophan assay. We used NanoDrop 2000 at 280 nm.

### 3.3. Trypsin Digestion and Peptide Clean-Up for LC-MS/MS

Our group prepared the samples as per Wisniewski et al.’s [22] protocol with minor modifications.

16.Remove the supernatant and replace it with 200 μL of lysis buffer (8 M urea in 100 mM ammonium bicarbonate with a final concentration of DTT of 50 mM), followed by incubation at 20 °C for 1 h.17.Put the Vivacon ultrafiltration spin columns’ filters into the provided 1.5 mL centrifuge tube and rinse the columns three times with 100 μL of 100 mM ammonium bicarbonate by centrifuging at 14,000× *g* for 10 min, followed three times with 100 µL of lysis buffer. Critical Step: Once the columns are equilibrated, try not to dry them out. This can be achieved by retaining a small amount of buffer on top of the column.18.Replace the collection tube with a fresh 1.5 mL centrifuge tube. Load the tissue samples into the spin columns and centrifuge them at 14,000× *g* for 10 min at room temperature. Critical Step: To make sure the proper binding of proteins to the Vivacon ultrafiltration spin column membrane, this step can be repeated twice with the flow-through.19.Discard the flow-through and subsequently alkylate the samples with 55 mM IAA in 100 mM ammonium bicarbonate, followed by incubation in the dark at room temperature for 30 min.20.Centrifuge the filters as in step 17 above and wash the membrane twice with 100 μL of 100 mM ammonium bicarbonate, followed by one wash with 100 μL of 50 mM ammonium bicarbonate.21.Digest the alkylated proteins at 37 °C overnight with sequencing grade trypsin at an enzyme-to-substrate ratio of 1:50 in 100 μL of 5 mM ammonium bicarbonate. Critical step: To minimize the evaporation and drying of the peptides, the bottom of the tubes can be filled with water and must be sealed using parafilm.22.After overnight incubation, add 0.1% FA to stop the digestion.23.Elute tryptic peptides off the filter by centrifuging the tubes at 14,000× g for 10 min.24.Dry the samples by vacuum centrifugation without heating. Pause step: Dried samples can be stored for months at −80 °C.25.Reconstitute the peptides in 2% (*v*/*v*) can, sonicate for 5 min, vortex, and centrifuge briefly.26.Estimate the peptide concentration using a nanodrop at 205 nM wavelength.27.Acidify the samples to a final concentration of 0.1% FA and place them in HPLC vials for peptide LC-MS/MS analysis (proceed to step 29). Alternatively, an optional extra clean-up step can be performed by using C18 Zip-tips. This will remove any residual salt that may be remaining.28.C18 ZipTip preparation: Prepare the ZipTips as per the manufacturer’s protocol. Critical step: Use one tip per sample. Equilibrate the tips three times with 100% can followed by three times with 0.1% FA (*v*/*v*).Load the peptide samples ten times by pipetting the digest up and down.Wash the samples six times with 0.1% FA (*v*/*v*).Elute the bound peptides six times with 60% (*v*/*v*) can containing 0.1% (*v*/*v*) FA.Dry the eluted peptide mixture using a speedVac without heating.Resuspend the peptides in 2% (*v*/*can*ACN, sonicate for 5 min, vortex, and centrifuge briefly. Pause step: Samples can be stored at −80 °C until required.

### 3.4. Nano-LC-MS/MS Analysis

Set up the LC and MS instrument as described under the equipment setup.

29.Prepared peptide samples can now be loaded onto an HPLC system connected online to a mass spectrometry. Our laboratory uses an ultimate 3000 HPLC system (Dionex, Thermo-Fisher Scientific) connected online to a maxis impact II QTOF (Bruker Daltonics).30.Inject ~1 µg of each sample and load for 8 min at a flow rate of 5 µL/min in 2% (*v*/*can*ACN, 0.1% (*v*/*v*) FA onto a C18 trapping column. Critical step: Based on the flow rate and tubing size, the sample loading time and gradient should be optimized for the instrument in use.31.Separate the peptides onto a C18 analytical column using 2% can*v*) ACN in 0.1% (*v*/*v*) FA (Buffer A) and 80% (*v*/*v*) ACN in 0.1% (*v*/*v*) FA (Buffer B) at a flow rate of 300 nL/min. The gradient used in our laboratory was 5% to 45% buffer B over 130 min, followed by a gradual increase in buffer B from 45% to 90% for 1 min and then held at 90% buffer B for 20 min, followed by re-equilibration of the column for 20 min at 5% buffer B, for a total run time of 180 min.32.Depending on the selected ion source and the HPLC flow rate, the appropriate settings for the nebulizer and dry gas must be selected. In our laboratory, the column eluent from the LC was connected online to Bruker’s CaptiveSprayTM source optimized to a capillary voltage of 1300 V with capillary temperature of 150 °C, at a nebulizing pressure of 3 L/mins.33.Data-dependent acquisition can be performed using Bruker’s Shotgun InstantExpertiseTM method. This is an advanced intelligent method to provide greater coverage and data analysis compatibility. Instead of the typical Top Intensity or Increasing *m*/*z* selection, this method uses a simplified acquisition model via an intelligent duty cycle optimizer. In this method,The depth of sampling is auto-optimized;The acquisition speed for each MS/MS event is auto-regulated;The acquisition exclusion is auto-optimized;The peak depth is optimized on the collision cell recovery using factory settings.34.On the orthogonal TOF systems, the mass range for the MS and MS/MS mode is the same. For tryptic peptides, we recommend setting the mass range to 50 to 2200 *m*/*z*.35.Exclude the singly charged precursor ions from acquisition, which prevents the selection of singly charged background ions, as well as polymer signals.36.As determined by the *m*/*z* of the precursor ion, the collision energy ranges from 23 to 65%.37.Set up a sample table in HyStar and select the appropriate LC and MS methods for the sample analysis.

### 3.5. LC-MS/MS Data Analysis

For quantitative label-free proteomics, LC-MS/MS raw files can be directly analyzed using freely available MaxQuant software at https://www.maxquant.org/download_asset/maxquant/latest (accessed on 10 March 2021).

Detailed steps on how to run maxquant with integrated Andromeda search engine is explained by Tyanove et al. (2016) [23]. Note: MaxQuant version 1.5.2.8 is used in this study.

38.The standard Bruker QTOF settings can be used with a mass error tolerance of 40 ppm; more specific parameters are defined as explained below:
Select the appropriate. fasta files, for, e.g., UniProt human-reviewed database;Select the digestion enzyme for Trypsin;Set the maximum number of missed cleavages to 1;Select fixed modification to carbamidomethyl of cysteines;Select the variable modification of oxidation of methionine;Set the protein false discovery rate (FDR) and peptide spectrum match FDRs to 1% using a target decoy approach;Set a minimum peptide length of seven amino acids;Set only unique and razor peptides when reporting protein identifications/

The mass spectrometry proteomics data were deposited in the ProteomeXchange Consortium via the PRIDE partner repository [24] with the data set identifier PXD018538.

### 3.6. In Situ Tryptic Peptide MALDI MSI Analysis

Prepare the slide as per Gustafsson et al. [25].

39.Place the ITO slide onto a heating block at 60 °C for 1 h.40.Wash the slide twice in 100% xylene for 5 min each.41.Wash the slide twice in 100% ethanol for 2 min each.42.Rinse the slide twice with 10 mM ammonium bicarbonate for 5 min each.43.Place the slide in an empty cleaned Coplin jar and fill the remaining slots with blank super frost slides (to prevent the formation of big air bubbles during the antigen retrieval step).44.Fill the Coplin jar with 10 mM citric acid monohydrate and microwave it at high power for 1.05 min or until the citric acid starts to boil.45.Once the citric acid reached the boiling point, microwave the slide for 10 min at power 10.46.Following microwave incubation, rapidly transfer the Coplin jar to a heating block set at 98 °C for 30 min.47.Remove the slide from the Coplin jar and allow it to cool down at room temperature.48.Rinse the slide twice with 10 mM ammonium bicarbonate.49.Dry the slide at room temperature for ~10 min.50.Apply teach marks on the 4 sides of the slide using Tipp-Ex (water-based white out).51.Scan the slide at a 2400 dpi resolution (CanoScan 5600 F, Canon, Thailand).52.Adjust the spray offset on the ImagePrep station (Bruker Daltonics), such that the spray lasts for at least 54 sec, and cover the whole slide slot. Set the Global Power Adjustment at 38% spray power with 0% modulation.53.Dilute 40 µL of the trypsin gold aliquot with 160µL of 25 mM ammonium bicarbonate and load directly onto the spray generator of the ImagePrep.54.Spray the trypsin using the trypsin deposition method as detailed in 6.2.1.55.After trypsin deposition, incubate the slide at 37 °C for 2 h in a humidified chamber.56.After 1.45 h, thaw, vortex, and quickly pulsecentrifuge an aliquot of internal calibrant and spray the internal calibrant using the same method as that of trypsin.57.Fill the ImagePrep solution vial with the matrix and adjust the spray offset for the matrix deposition as detailed in6.2.1.58.After matrix deposition, remove the slide from the Imageprep and clean both ends of the slides with 100% methanol.59.Load the prepared slide into an MTP slide adapter II and acquire the data using the ultrafleXtreme MALDI TOF/TOF instrument.60.Using the flexcontrol, create an autoXecute method with the following required settings:Select the optimized and calibrated flexControl method in the general tab.Turn the fuzzy control off and set the optimized laser power (we used 50% laser power).No background list should contain in the evaluation tab.Acquire 500 shots in 500 shot steps with dynamic termination and random walk-off.Choose an appropriate data processing method as in step 61.61.Create a flexAnalysis method with the following defined settings: smoothing by Gaussian (2 cycles with the width of *m*/*z* 0.02), TopHat baseline subtraction, Monoisotopic SNAP peak algorithm, and the quadratic recalibration using the calibrations masses (1296.685, 1570.677, 2147.19,9 and 2932.588) with a mass tolerance of 500 ppm.62.Open flexImaging and create a new sequence with a sample preparation type of uniformly distributed coating and 60 µm raster width for this workflow.63.Teach the slide using the scanned slide and check the teaching by moving the sample carrier at various points on the slide.64.Create the polygon measurement of the region(s) and save the flexImaging sequence.65.Optimize the laser power by shooting the laser outside the tissue region and on the calibrants sprayed.66.Calibrate the flexcontrol method from the calibration tab in the flexcontrol and save the method. Critical Note: Make sure monoisotopic peaks are resolved to baseline.67.Start the data acquisition from the flexImaging checklist menu.68.After data acquisition, remove the slide from the instrument and wash the matrix using 70% ethanol (AR grade) and H&E stain using the standard procedure; briefly,Rinse the slide with 70% ethanol (AR grade) for 5 min or until the matrix is removed;Dip into deionized water for 30 s;Immerse in hematoxylin solution (Mayer’s) for 50 s;Wash with tap water for 5 min;Immerse in eosin solution for 30 s;Rinse with 70% ethanol (AR grade) for 30 s;Rinse with 90% ethanol (AR grade) for 30 s;Rinse with 100% ethanol (AR grade) for 30 s;Rinse with 100% isopropanol (AR grade) for 30 s;Rinse twice with xylene for 1 min each;Apply the mounting medium onto the cover slide;Cover the tissue section with the coverslip and gently pressed it onto it;Let the slide dry overnight at room temperature under the fume hood.69.Scan the slide at a 20× objective using a Nanozoomer and co-register the scanned slide using flexImaging edit drop-down menu “co-register image” and save the sequence.70.Mark the region(s) of interest as per the pathologist’s annotation, in our case EIC and normal endometrium.71.Load the raw data into SCiLS lab v2016b and process them there by selecting the instrument type “Time-of-flight”, Import raw, fallback to reduced spectra, and TopHat as the baseline subtraction method.72.Open the imported dataset, set the denoising as weak, baseline removal as TopHat, and change the file properties from the file’s drop-down menu to 0.125 Da.73.Using the Peak picked algorithm from the tools drop-down menu, create a list of peaks detected and manually go through each *m*/*z* value that is differentially regulated in the region(s) of interest (Figure 3).

## 4. Expected Results

In this manuscript, we outline two mass spectrometry protocols, which complement each other and provide unique insight into the molecular details of concurrent ovarian and endometrial PLs from a single patient.

The EIC and STIC lesions were found in the endometrium and fallopian tube, respectively, and tissues were embedded via a standard FFPE protocol [19]. The identification of the PL was achieved through immunoperoxidase staining for p53 and MIB1 (data not shown), which are identifying features of these PLs [7,26]. In addition, the indications of cell transformation, such as hyperchromasia, nuclear atypia, and nucleomegaly, were utilized to advise an experienced pathologist in the identification of precancerous regions [7,27] (Figure 2).

Using MALDI-MSI, we identified almost 600 molecular features (Appendix A), of which 72 could be used to distinguish between healthy and precancerous tissue. We chose to display the analyte with a mass-to-charge ratio (*m/z*) of 1220.64 Daltons (Da) (Figure 3), which, as demonstrated in the ROC curve, can be utilized to confidently delineate between cancerous and non-cancerous regions in this sample. While the analyte *m/z* 1120.64 was seen to be significantly enriched in precancerous regions than benign regions, it is heterogeneously distributed within the precancerous tissue. This could be due to several factors, including the heterogenous mix of cells and connective tissue within the section or heterogeneous distribution of this analyte within and between cells. Further investigation into the nature of this analyte would be required to reveal this; however, it still serves as a clear discriminating factor between precancerous and benign tissue. While antibody-based techniques are a powerful tool for identifying different regions of tissue, they can be complemented by the MALDI MSI method presented here to gain additional molecular information.

While MALDI MSI is a powerful technique to gather spatially defined molecular information, the lack of peptide fragmentation makes protein identification difficult. To overcome this, we incorporated LC-MS/MS to complement our MALDI MSI analysis (Figure 1).

To excise regions of interest for LC-MS/MS analysis, we employed LCM using a Leica Zeiss LCM system. Advised by the annotations of an experienced pathologist, which show the altered distribution of analytes as visualized through MALDI MSI, we were able to successfully isolate these small, but biologically relevant, regions of tissue.

Proteins were extracted from the isolated regions of tissue using an antigen retrieval protocol [25] and digested using a modified FASP protocol [22] and then analyzed on maxis impact II QTOF connected online to an ultimate 3000 nano-LC before protein identification via MaxQuant. Through the isolation of peptides, subsequent fragmentation, and analysis of these secondary MS spectra, the details of amino acid structures were gathered. Protein identification was achieved using the MaxQuant analysis platform in combination with the Andromeda database.

This resulted in a total of 453 proteins detected across the four tissue types [21]. Through the analysis of these data, several metastasis-related proteins were identified [21]. For example, the protein transketolase was identified here in both the STIC and EIC precancerous lesions. This protein has previously been seen to be upregulated in peritoneal metastasis of ovarian cancer and its expression correlates with reduced overall survival [28]. The presence of metastasis-related proteins in the PLs investigated here merits further investigation into their migratory and metastatic potential and the molecular events that underpin them.

We have previously utilized this approach to investigate spatially defined alterations in protein expression in vulvar and endometrial cancer [29,30]. These techniques have also been utilized to accurately classify endometrial cancers with associated lymph node metastasis [31].

Here, we have outlined a method for the complementary tissue-specific proteomic analysis of ovarian and endometrial PLs. MALDI MSI investigation of these tissues was able to identify an analyte that could successfully delineate between cancerous and non-cancerous tissue (Figure 3). Guided by MALDI MSI ion intensity maps and the annotation of an experienced pathologist, we were able to dissect precancerous areas using LCM (Appendix A). We were able to successfully identify hundreds of proteins from each sample using LC-MS/MS. Our lab has previously utilized these techniques [2,18,21,25,30,31,32,33,34,35,36] demonstrating their broad utility.

Depending on sufficient sample material and utilization of the appropriate data acquisition method, LC-MS/MS also has the potential to quantify relative protein abundance between ROIs or between samples. Due to the unique nature of the samples investigated, this was not feasible here. Though the small sample size is a limitation, which could be overcome by the analysis of numerous biological replicates, LC-MS/MS can achieve relative quantification of protein abundance. This could provide further information about molecular perturbations present in PL compared to their neighboring tissues.

## 5. Discussion

In this manuscript, we outline a protocol for the acquisition of spatially defined proteomic information from small regions of FFPE tissue. This was achieved through the application of MALDI MSI to acquire spectra from across the tissue section in a spatially defined manner. Annotation by an experienced pathologist, which showed the altered distribution of analytes as visualized through MALDI MSI, was used to advise LCM or regions of interest, which were then investigated through a specific sample preparation workflow coupled with high-sensitivity label-free LC-MS/MS analysis. This analysis resulted in the identification of hundreds of proteins differentially expressed between the different tissues, including some that were indicative of a metastatic and migratory potential possessed by these PLs.

The identification and characterization of the PLs investigated here are made difficult by their small size, particularly in relation to the surrounding healthy tissue. Here, we demonstrate that MALDI MSI is a technique with great utility for identifying and characterizing small tissue regions from a larger tissue section. This technique utilizes a matrix that is sprayed homogeneously over the tissue, facilitating laser-induced ionization/desorption of analytes, which are then analyzed through a time-of-flight (TOF) mass spectrometer. This provides spectra at each discrete location that the laser was directed towards. By acquiring data from these locations across the tissue, a spatially defined molecular analysis can be achieved.

To demonstrate the utility of MALDI MSI, we presented the distribution of a single ion (*m/z* = 1220.64 Da) that was able to differentiate between the healthy and precancerous areas of the tissue. In combination with the numerous other analytes present in this analysis (Appendix A), this shows that the proteomic landscape can be significantly different between adjacent regions of tissue and that PLs represent a significant perturbation in their protein expression compared to the tissue from which they derived. Our lab has previously used MALDI MSI to delineate regions of ovarian cancer tissue [34] to investigate spatially defined alterations in protein expression in the vulva and endometrial cancer [29,30] and to provide predictive information on the metastatic status of endometrial cancer based on a primary cancer biopsy [31], demonstrating the robustness and broad utility of this technique.

Here, we performed MALDI MSI and LC-MSMS on consecutive tissue sections to acquire complementary data on analytes and their special distribution. A recent study by Mezger et al. (2021) demonstrated the application of MALDI MSI followed by LCM and LC-MSMS on a single tissue section with only minor impacts on protein identification [37]. This merits further investigation in the context of PLs to reduce inconsistencies that could arise through the use of consecutive sections.

Through the application of tailored sample preparation and label-free LC-MS/MS analysis, we demonstrate the ability to acquire detailed proteomic information from small formalin-fixed tissue sections. This was facilitated through the precise isolation of regions of interest through LCM. This technique is well-established and holds great utility for the molecular investigation of small but biologically important tissue regions. In combination with the accurate annotation and characterization of different tissue regions, this can facilitate the LC-MS/MS-mediated identification of proteins, thereby providing a temporal snapshot into the molecular landscape within these samples.

Further analysis of the proteins identified in this experiment has revealed several proteins indicative of a migratory phenotype in these PLs, including transketolase. This was achieved through a simple analysis of proteins identified within the cancerous regions that were not present in the healthy tissue. A larger sample set would provide the opportunity to acquire quantitative information regarding relative protein abundance between samples. This has the potential to characterize the up- and down-regulation of proteins, which is indicative of oncogenic transformation or migratory potential.

A larger sample set would also facilitate more complex data analysis approaches including, but not limited to, network and pathway analysis. A further advantage of this approach is that it investigates the tissue in an unbiased manner, facilitating the discovery of new molecular features rather than relying on targeted approaches such as immunohistochemistry.

One of the limitations of the proteomics analysis of small sample sizes is that many proteins are extracted at levels that are below the limit of detection for this technique. However, with the improvement in instrumentation, this proteome coverage will continue to improve rapidly [38].

While the technology is advancing, there is the potential to improve protein coverage by using complementary proteomics approaches. One of the advantages of this protocol is that the antigen retrieval and LCM processes can be coupled with a range of other MS-based techniques using method-specific sample preparation steps. For example, our lab has investigated spatially defined glycan expression through a combination of MALDI mass spectrometry imaging, LCM, glycan-specific sample preparation, and LC-MS/MS, in a range of sample types and conditions [32,33,34,35,36]. This opens the possibility of future systematic investigations of precancerous samples through a range of omics techniques that could unlock more molecular details.

PLs located within the endometrium and fallopian tube require more detailed investigation to understand the early stages of their oncogenic development. Barriers to this investigation include the acquisition of the PLs in the first instance, accurate identification of precancerous regions, and the acquisition of unbiased spatially defined and detailed molecular information. There are several challenges relating to the acquisition of PLs in the first instance, but the method presented here overcomes barriers to the acquisition of spatially defined and detailed molecular information from these samples. This is achieved through MALDI MSI and LCM followed by LC-MS/MS. In this pilot study, these complementary techniques provide a detailed picture of the molecular terrain of endometrial and fallopian tube tissue and suggest that the PLs located there possess a migratory phenotype that merits further investigation. In conjunction with other mass spectrometry-based methods, the workflow described here can be used to identify and quantify proteins from small areas of tissue and provide novel insights into the early stages of cancer development, which are complementary to genomics data.

## 6. Reagent and Equipment Set-Up

### 6.1. Reagent Set Up


**10 mM citric acid monohydrate pH 6**


For 500 mL of 10 mM citric acid, weigh 1.05 g of citric acid monohydrate and dissolve in 480 mL milli-Q^®^ water. Adjust the pH to 6.0 using 1 M NaOH (~16 mL required). Adjust the volume to 500 mL with milli-Q^®^ water. Note: It can be stored for 30 days at room temperature. It can be stored for 30 days at room temperature.


**100 mM ammonium bicarbonate**


For 50 mL of 10 mM ammonium bicarbonate, weigh 394 mg into a polypropylene container and adjust the volume to 50 mL with milli-Q^®^ water. Note: Must be prepared fresh, as the pH of the solution changes over time.


**10 mM ammonium bicarbonate**


For ~500 mL of 10 mM ammonium bicarbonate, dilute 50 mL of 100 mM ammonium bicarbonate into 450 mL milli-Q^®^ water. Note: Must be prepared fresh, as the pH of the solution changes over time.


**25 mM ammonium bicarbonate**


For ~500 µL of 25 mM ammonium bicarbonate, dilute 125 µL of 100 mM ammonium bicarbonate into 475 µL milli-Q^®^ water. Note: Must be prepared fresh, as the pH of the solution changes over time.


**8 M urea in 100 mM ammonium bicarbonate**


For 10 mL of 8 M urea in 100 mM ammonium bicarbonate, weigh 4.80 g urea and 80 mg ammonium bicarbonate in a polypropylene container and adjust the volume to 10 mL with milli-Q^®^ water. Note: Use high-quality reagents and make fresh each day.


**DTT (1,4-Dithiothreitol)**


Must be made fresh or from a concentrated solution stored at −80 °C. We make it at 1 M and store it in 100 µL aliquot at −80 °C.


**IAA (Iodoacetamide)**


Must be made fresh or from a concentrated, frozen solution. We make it at 550 mM and dilute it 1/10 before use with 8 M Urea in 100 mM ammonium bicarbonate. Note: IAA is light-sensitive.


**LC buffer A (2% (*v*/*v*) ACN with 0.1% (*v*/*v*) FA)**


Dilute 20 mL of ACN in 979 mL of milli-Q^®^ water and 1 mL of FA.


**LC buffer B (80% (*v*/*v*) ACN with 0.1% (*v*/*v*) FA)**


Dilute 800 mL of ACN in 199 mL of milli-Q^®^ water and 1 mL of FA.


**Trypsin Gold**


Dissolve 100 µg lyophilized trypsin gold in 200 µL 5 mM NH_4_HCO_3_ and split it into 40 µL aliquots and freeze at −80 °C until required.


**Internal calibrant solution**


Prepare the calibrant stock solution as per Gustafsson et al. (2012) [39]. Briefly, combine the below calibrants (Table 1) in appropriate amounts, vortex, and quickly pulsecentrifuge. Split into 100 µL aliquots and store at −80°C until required.


**7 mg/mL α-cyano-4-hydroxycinnamic acid**


Dissolve 70 mg of HCCA in 10 mL of 50% ACN, 49.8% milli-Q^®^ water and 0.2% TFA.

### 6.2. Equipment Set-Up

#### 6.2.1. Nano-HPLC System

The LC instrument needs to be optimized in advance. The instrument-specific settings utilized in our lab are as follows:


**Analytical column**


Acclaim PepMap100 C18 75 μm × 50 cm, 164,568, Thermo-Fisher Scientific


**Trap column**


Acclaim PepMap100 C18 75 μm × 20 mm, 164,535, Thermo-Fisher Scientific


**Mobile phases**


A: 2% (*v*/*v*) ACN with 0.1% (*v*/*v*) FA in milli-Q^®^ water,

B: 80% (*v*/*v*) ACN with 0.1% (*v*/*v*) FA in milli-Q^®^ water


**Loading solvent**


2% (*v*/*v*) ACN with 0.1% (*v*/*v*) FA in milli-Q^®^ water


**Flow rate**


300 nL/min


**Gradient**


5% buffer B for 8 min

5–45% buffer B over 130 min

45–90% buffer B in 1 min

90% buffer B for 20 min

90–5% buffer B for 1 min

5% buffer B for 20 min


**Column temperature**


60 °C


**Sample loading**


µL pickup


**Injection amount**


~1 µg


**ImagePrep**


Adjust the spray offset for trypsin and matrix deposition prior to the spray.
-Deposit the trypsin using Bruker’s default method with minor modifications. Settings used:38% Spray Power with 0% Modulation, total number of 30 cycles with spraying time of 1.25 s, and 45 s of drying.-Deposit the matrix using Bruker’s default HCCA method with minor modifications. Settings used (Table 2):

#### 6.2.2. Mass Spectrometer System


**ESI-maxis Impact II QTOF**


Mass spectrometer (MS) must be calibrated as per the vendor’s recommendations. For example, calibration on maxis Impact II QTOF can be performed using Agilent’s ESI-L low concentration tuning mix on an enhanced quadratic calibration mode using Bruker’s default calibration MS method.

Additionally, optimize the ionizing and spray condition and, if necessary, adjust the MS/MS timing with regard to the chromatographic peak width, mean number of precursor ions, averages in MS & MS/MS as well as active exclusion timing.


**MALDI-ultrafleXtreme TOF/TOF**


On the flexControl, load an appropriate reflectron-positive method for the *m/z* range of 800–4500.

Optimize the laser power, offset, repetition rate, detector gain, and acquisition rate.

## Figures and Tables

**Figure 1 mps-05-00057-f001:**
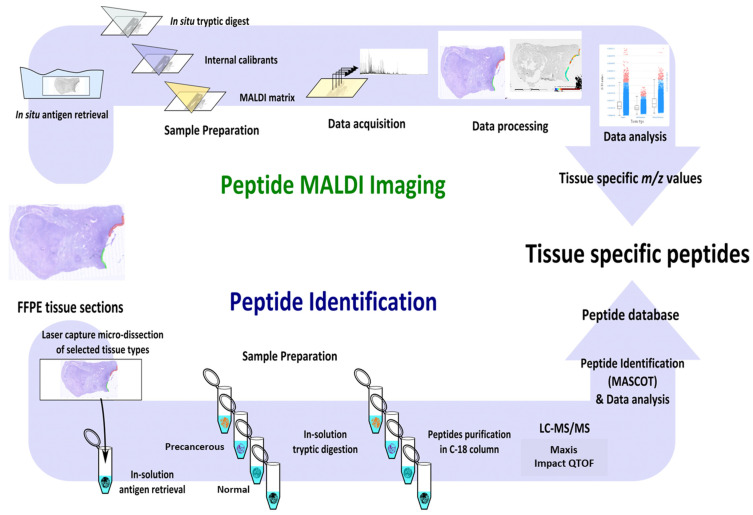
(Adapted with permission from from G. Arentz et al. (2017) [18]): Workflow of proteomic analysis of small precancerous regions extracted from FFPE tissue. Peptide MALDI Imaging:MALDI MSI workflow to acquire spatially defined molecular information. Peptide Identification: LCM extraction of precancerous regions followed by LC-MS/MS analysis for the identification of proteins differentially expressed between healthy and precancerous tissue regions.

**Figure 2 mps-05-00057-f002:**
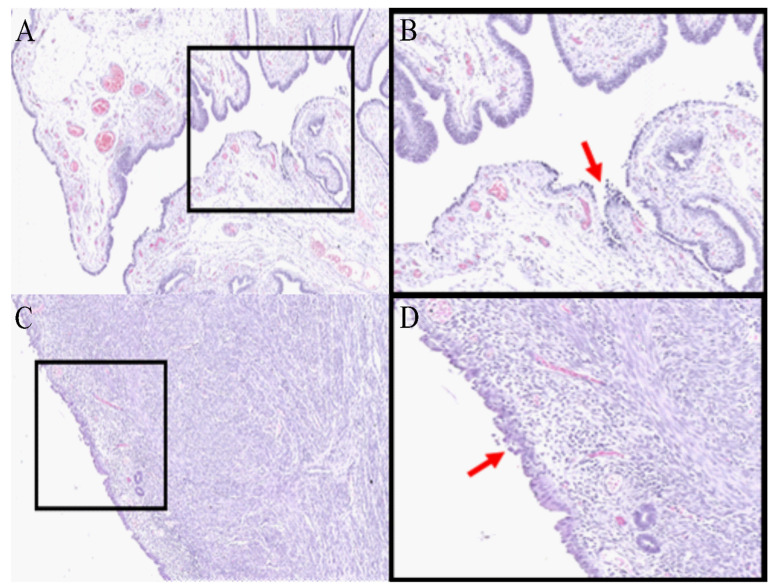
Hematoxylin and eosin-stained fallopian tube (**A**,**B**) and endometrium tissue (**C**,**D**) at 6× (**A**,**C**) and 12× (**B**,**D**) magnification. Areas of STIC (**B**) and EIC (**D**) are indicated by the red arrows. (Figure adapted, with permission, from “Proteomics Analysis of Serous Lesions of the Endometrium and Fallopian Tube Reveals Their Metastatic Potential”, Acland et al. (2020) [21].

**Figure 3 mps-05-00057-f003:**
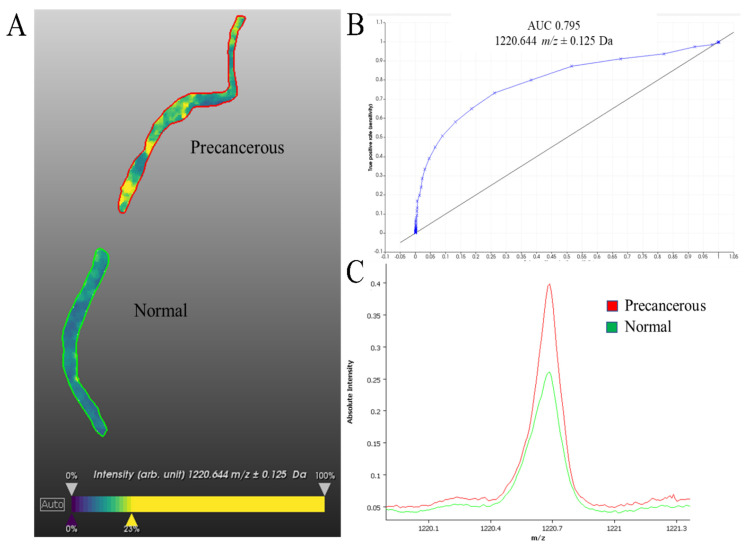
MALDI MSI on an endometrial tissue (**A**) Representative ion intensity image of *m/z* 1220.644 ± 0.125 Da. (**B**) The ROC curve of *m/z* 1220.644 (AUC 0.795) for precancerous versus normal region. (**C**) Comparative spectra of *m/z* 1220.644 Da between precancerous and normal region. Scale bar is 1.4 cm, ion intensity ranges from blue (lowest) to yellow (highest).

**Table 1 mps-05-00057-t001:** Details of internal calibrants utilized in the set up of the MALDI UltrafleXtreme TOF/TOF.

Calibrants	Peptide Mass [M + H]^+^	Final Concentration
**Angiotensin I34-43 (C10-00002, BioRad, Hercules, CA, USA)**	1296.685	0.4 pmol/µL
**[Glu1]-fibrinopeptide B (F3261, Sigma-Aldrich, St. Louis, MI, USA)**	1570.677	0.4 pmol/µL
**Dynorphin A 1-17 (2032, Auspep, Tullamarine, Victoria, Australia)**	2147.199	2.0 pmol/µL
**ACTH 1-24 (A0298, Sigma-Aldrich, St. Louis, MI, USA)**	2932.588	2.0 pmol/µL
**TFA (10%)**		0.2% (*v*/*v*)
**Milli-Q^®^ Water (≥18.2 MΩ)**		

**Table 2 mps-05-00057-t002:** Settings used for deposition of HCCA matrix with the ImagePrep instrument.

Phase	Sensor	Nebulization	Incubation	Drying
**1**	0.65 V within 8–20 cycles	20% spray power ±35% modulation with fixed spray time of 2.5 s	10 s	90 s
**2**	30 s drying			
**3**	0.1 V within 4–10 cycles	20% spray power ± 35% modulation with 0.05 V sensor-controlled spray time	30 s ± 30 s	Complete dry every cycle, safe dry 10 s
**4**	0.1 V within 8–12 cycles	20% spray power ± 35% modulation with 0.1 V sensor-controlled spray time		Grade 20 ± 40% complete dry every 2 cycle, safe dry 20 s
**5**	0.3 V within 12–30 cycles	25% spray power ± 35% modulation with 0.2 V sensor-controlled spray time		Grade 30 ± 40% complete dry every 3 cycle, safe dry 30 s
**6**	0.6 ± 0.5 V within 20–64 cycles	25% spray power ± 35% modulation with 0.3 V sensor-controlled spray time		Grade 40 ± 40% complete dry every 4 cycle, safe dry 40 s
**7**	0.6 ± 0.5 V within 20–64 cycles	25% spray power ± 35% modulation with 0.3 V sensor-controlled spray time		Grade 40 ± 40%, complete dry every 4 cycle, safe dry 40 s

## Data Availability

The mass spectrometry proteomics data have been deposited to the ProteomeXchange Consortium via the PRIDE partner repository [24] with the data set identifier PXD018538.

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
