# Peer review of "A Protocol for the Acquisition of Comprehensive Proteomics Data from Single Cases Using Formalin-Fixed Paraffin Embedded Sections"

_mps, 2022, doi:10.3390/mps5040057_

Round 1

Reviewer 1 Report

In the Protocol manuscript entitled “A protocol for the acquisition of comprehensive proteomics data from single formalin-fixed paraffin embedded sections”, Acland et al. present a workflow which allows spatial proteomics data obtained by MALDI-MSI to be used as a guide for Laser Capture Microdissection (LMD) and subsequently complemented with nLC-MS/MS proteomics on single FFPE tissue sections. On the whole, the manuscript is well written and much of the methodology and technical aspects are written in a sound, clear, and comprehensive manner. Moreover, the possibility to extract more in-depth spatial omics information from a single FFPE tissue section is a particularly pertinent topic, especially in the context of rare diseases. However, here are number of critical aspects which hinder the suitability of the manuscript in its current form and must be addressed:

1) From what I can gather, two FFPE tissue sections are required to complete this protocol. Given that MALDI-MSI is inherently performed on a single tissue section, I would argue that the title is particularly misleading and using two tissue sections, one for MS-imaging and the second for traditional proteomics is not particularly novel.

2) This is hampered by the fact that it is not clearly described how the MALDI-MSI data was used to guide the LMD. Were the co-ordinates transferred to the LMD precisely? Was this performed in a broad manner? This should be more clearly described and better highlighted the regions tha

3) In Figure 3, a H&E counterpart of the imaged tissue sections should be added in order for the reader to understand the added value of considering the MALDI-MSI dataset to guide LMD. Does it also highlight additional regions that are not underlined by traditional histology? Moreover, the tissue distribution of the protein feature at m/z 1120.64 is not homogenous within the precancerous tissue. It would also be better to justify this heterogeneous distribution with the aid of the H&E image here.

4) Regarding the novel aspects, there are recent works which also highlight the possibility to use MALD-MSI guided spatial proteomics on a single tissue section (Mezger et al; https://doi.org/10.1021/acs.analchem.0c04572). Whilst the author is aware that this was using spatial spatial metabolomics of FFPE tissue to guide the LMD, it is also technically possible to use the spatial proteomics data to guide the LMD on a single FFPE tissue section. Thus, it would be nice to see the authors stress the advantage of the presented workflow.

5) The authors state that “Using MALDI MSI, we identified almost 600 hundred molecular features (Supplementary Table 1) of which 72 could be used to distinguish between healthy and precancerous tissue.”. Considering that the nLC-MS/MS analysis is designed to complement this, and protein annotation is arguably the biggest limitation with MALDI-MSI, how many of these protein features could be assigned a putative identity? Highlighting this information would better stress the harmony between the two approaches and support the statement “we can improve protein coverage by using complementary proteomics approaches” present on line 534, especially considering that the “MALDI-MSI guide” is not well documented in the current form.

I hope that the authors will be able to address these issues. Despite the significant lack of novelty, we should consider that the recent works regarding MALDI-MS imaging guided spatial omics is performed on instrumentation that may not be available to many users working in the field. Thus, the reviewer feels that if the major changes are made to this Protocol manuscript then it could represent a workflow that renders it more accessible to a wider audience.

Author Response

In the Protocol manuscript entitled “A protocol for the acquisition of comprehensive proteomics data from single formalin-fixed paraffin embedded sections”, Acland et al. present a workflow which allows spatial proteomics data obtained by MALDI-MSI to be used as a guide for Laser Capture Microdissection (LMD) and subsequently complemented with nLC-MS/MS proteomics on single FFPE tissue sections. On the whole, the manuscript is well written and much of the methodology and technical aspects are written in a sound, clear, and comprehensive manner. Moreover, the possibility to extract more in-depth spatial omics information from a single FFPE tissue section is a particularly pertinent topic, especially in the context of rare diseases. However, here are number of critical aspects which hinder the suitability of the manuscript in its current form and must be addressed:

1) From what I can gather, two FFPE tissue sections are required to complete this protocol. Given that MALDI-MSI is inherently performed on a single tissue section, I would argue that the title is particularly misleading and using two tissue sections, one for MS-imaging and the second for traditional proteomics is not particularly novel.

The reviewer is correct, we need two consecutive sections of the tissue, one on the PEN membrane slide to perform the LMD and one on an ITO slide to perform MALDI-MSI. We did not intend to mislead the reader or reviewer with the title, but now realise that it needs to be more concise. We therefore propose to change the title to: A protocol for the acquisition of comprehensive proteomics data from single cases using formalin-fixed paraffin embedded tissue. (this change is reflected in the updated manuscript

2) This is hampered by the fact that it is not clearly described how the MALDI-MSI data was used to guide the LMD. Were the co-ordinates transferred to the LMD precisely? Was this performed in a broad manner? This should be more clearly described and better highlighted the regions tha

The LMD was mainly guided by the annotation of the pathologist. However, the image of the immunohistochemistry stained section was co-registered with the MALDI-MSI data and has confirmed alteration of proteins in the area annotated by the pathologist.

To clarify this, we have slightly changed the wording in lines 450 and 493.

3) In Figure 3, a H&E counterpart of the imaged tissue sections should be added in order for the reader to understand the added value of considering the MALDI-MSI dataset to guide LMD. Does it also highlight additional regions that are not underlined by traditional histology? Moreover, the tissue distribution of the protein feature at m/z 1120.64 is not homogenous within the precancerous tissue. It would also be better to justify this heterogeneous distribution with the aid of the H&E image here.

When we have analysed the sections using MALDI-MSI, data from the whole tissue section were acquired. The Scils software can be used to a visual analysis of the data, identifying regions that are distinct from each other. However, the regions of interest are very small, and the unsupervised approach did not provide separation of the areas of interest, mainly because of the disproportionate high number of spectra acquired outside the area of interest. We therefore compared two regions of similar size and therefore similar number of spectra instead. With this targeted approach we were unable to identify any additional regions of interest. Uneven tissue distribution of m/z of interest is common in MALDI-MSI and could be caused by ion suppression. Therefore we compare regions instead of individual mass spectra.

We have clarified this in the text in line 435

4) Regarding the novel aspects, there are recent works which also highlight the possibility to use MALD-MSI guided spatial proteomics on a single tissue section (Mezger et al; https://doi.org/10.1021/acs.analchem.0c04572). Whilst the author is aware that this was using spatial spatial metabolomics of FFPE tissue to guide the LMD, it is also technically possible to use the spatial proteomics data to guide the LMD on a single FFPE tissue section. Thus, it would be nice to see the authors stress the advantage of the presented workflow.

As outlined above, we primarily utilized pathologist annotations for the identification of regions of interest. This annotation was supported by the MALDI MSI data rather than being wholly directed by it. Regardless, the publication referred to by the reviewer is of interest and we have added a section in the discussion to reflect this (line 520)

5) The authors state that “Using MALDI MSI, we identified almost 600 hundred molecular features (Supplementary Table 1) of which 72 could be used to distinguish between healthy and precancerous tissue.”. Considering that the nLC-MS/MS analysis is designed to complement this, and protein annotation is arguably the biggest limitation with MALDI-MSI, how many of these protein features could be assigned a putative identity? Highlighting this information would better stress the harmony between the two approaches and support the statement “we can improve protein coverage by using complementary proteomics approaches” present on line 534, especially considering that the “MALDI-MSI guide” is not well documented in the current form.

We usually don’t try identify the putative proteins where each individual peptide came from and instead focus on the ones that are differentially abundant across the regions of interest. We use LC-MS/MS data generated from the same tissue to get a putative identification of the protein of interest. However, in this case we did not try to identify the proteins.

We have altered the sentence in line 550 to clarify this.

I hope that the authors will be able to address these issues. Despite the significant lack of novelty, we should consider that the recent works regarding MALDI-MS imaging guided spatial omics is performed on instrumentation that may not be available to many users working in the field. Thus, the reviewer feels that if the major changes are made to this Protocol manuscript then it could represent a workflow that renders it more accessible to a wider audience.

Reviewer 2 Report

The manuscript by Acland et al. describes protocols for performing two types of mass spectrometry analyses on formalin-fixed paraffin embedded tissue sections. The combination of MALDI imaging and LC-MS/MS proteomic analysis to look at small tissue samples is an important means to examine precious clinical samples. Overall, the protocol is clear and understandable. A few minor modifications would be helpful.

In the critical step of Step 10, should it be simply to add the buffer rather than saying it “can” be added? When there is an expectation of buffer in Step 12, the suggestion seems more like a recommendation.

2.       In Step 17, the naming for the ultrafiltration devices should be consistent between the list of equipment and within the protocol steps to help prevent confusion.

3.       In the critical step of Step 21, it is hard to visualize where the paraffin goes to prevent evaporation and drying. A picture would help clarify. Also, is this paraffin or parafilm?

4.       Is there a potential pause point in Step 24 also for those that choose not to desalt?

5.       In Step 56 and in Section 6.1, what is flick centrifuging?

6.       Supplementary Table 1 was not included with the manuscript, only Supplementary Figure 1.

Author Response

The manuscript by Acland et al. describes protocols for performing two types of mass spectrometry analyses on formalin-fixed paraffin embedded tissue sections. The combination of MALDI imaging and LC-MS/MS proteomic analysis to look at small tissue samples is an important means to examine precious clinical samples. Overall, the protocol is clear and understandable. A few minor modifications would be helpful.

-  In the critical step of Step 10, should it be simply to add the buffer rather than saying it “can” be added? When there is an expectation of buffer in Step 12, the suggestion seems more like a recommendation.

We thank the review for picking up this mistake. It has been corrected in line 210

  1. In Step 17, the naming for the ultrafiltration devices should be consistent between the list of equipment and within the protocol steps to help prevent confusion.

This has been corrected in lines 229 and 236

  1. In the critical step of Step 21, it is hard to visualize where the paraffin goes to prevent evaporation and drying. A picture would help clarify. Also, is this paraffin or parafilm?

We are unable to provide a picture at this time. The reviewer is correct in that it should be 'parafilm' and this has been corrected in line 247.

  1. Is there a potential pause point in Step 24 also for those that choose not to desalt?

This is a potential pause point and this has been noted in line 250

  1. In Step 56 and in Section 6.1, what is flick centrifuging?

We have changed this to 'quick pulse centrifuge' to clarify in lines 354 and 613

  1. Supplementary Table 1 was not included with the manuscript, only Supplementary Figure 1.

Supplementary Table 1 is to be uploaded with the re-submission

Round 2

Reviewer 1 Report

Dear authors,

Thank you for submitting your revise article which contain amendments in line with the previous comments provided. Whilst I still feel that it would be useful to assign a putative identity to those m/z features which could discriminate the various regions of interest, and would increase the interest to the readership, the article can now be considered suitable for publication and the ambiguous aspects have now been clarified.

Congratulations on the work.